



# Chemical evolution of secondary organic aerosol tracers during high PM$_{2.5}$ episodes at a suburban site in Hong Kong over 4 months of continuous measurement

Qiongqiong Wang[1], Shan Wang[2], Yuk Ying Cheng[1], Hanzhe Chen[2], Zijing Zhang[2], Jinjian Li[2], Dasa Gu[2], Zhe Wang[2], and Jian Zhen Yu[1,2]

[1]Department of Chemistry, The Hong Kong University of Science & Technology, Hong Kong, China.
[2]Division of Environment & Sustainability, The Hong Kong University of Science & Technology, Hong Kong, China.

*Correspondence to*: Jian Zhen Yu (jian.yu@ust.hk)

**Abstract.** Secondary organic aerosol (SOA) makes a sizable contribution to fine particulate matter (PM$_{2.5}$) pollution, especially during episodic hours. Past studies of SOA evolution at episode-scale mainly rely on measurements of bulk SOA mass and few studies probe individual SOA molecular tracers. In this study, we continuously monitored at a bihourly resolution SOA tracers specific to a few common volatile organic compound (VOC) precursors at a suburban site in Hong Kong for four-month from the end of Aug. to Dec. 2020. The SOA molecules include tracers for SOA derived from biomass burning emissions, monoaromatics, naphthalene/methylnaphthalenes, and three biogenic VOCs (i.e., isoprene, monoterpene and sesquiterpene). Generally, the SOA tracers showed regional characteristics for both anthropogenic and biogenic SOA, as well as the biomass burning-derived SOA. This work focused on the seasonal variation and evolution characteristics of SOA tracers during eleven city-wide PM$_{2.5}$ episodes, which are defined to be periods of PM$_{2.5}$ exceeding 35 μg/m$^3$ at three or more of the 15 general air quality monitoring stations cross the city. Mass increment ratios (MIR), calculated as the ratio of mass concentration between before and during an episode, were examined for individual species in each episode. During most episodes, the SOA tracers were enhanced in their concentrations (i.e., MIR>1) and maximum MIR values were in the range of 5.5-11.0 for SOA tracers of different precursors. Episodes on summer and fall days showed notably larger MIR values than those falling on winter days, indicating a higher importance of SOA to formation of summer/fall PM$_{2.5}$ episodes. Simultaneous monitoring of six tracers for isoprene SOA revealed the dominance of the low-NOx pathway in forming isoprene SOA in our study region. The multiple monoterpene SOA products suggested fresher SOA in winter, evidenced by an increased presence of the early generation products. The current study has shown by example the precursor-specific SOA chemical evolution characteristics during PM$_{2.5}$ episodes in different seasons. This study also suggests the necessity to apply the high time resolution organic marker measurement at multiple sites to fully capture the spatial heterogeneity of the haze pollution at the city scale.



## 1 Introduction

The past two decades have seen the air quality in China substantially improved following the implementation of a series of stringent emission controls. Long-term trend analysis suggested the clear reduction of fine particulate matter ($PM_{2.5}$) in major city clusters in China such as Beijing–Tianjin–Hebei (BTH), the Yangtze River Delta (YRD), and the Pearl River Delta (PRD) region (e.g., Wang et al., 2016, 2020). However, the short-term PM episodic events are still frequently observed in recent years, especially in fall and winter. Short-term exposure to high levels of $PM_{2.5}$ during the episodes is associated with adverse

health effects, especially for the sensitive subgroups (Zanobetti et al., 2000). The causes of high PM episodes differ from one geographical location to another, due to location-specific emission sources and meteorological conditions. One needs to acquire knowledge about the chemical compositions and formation mechanism of the PM during episodes for formulating location-specific effective PM control measures.

Hong Kong (HK), with an area of about 1100 $km^2$, is located at the tip of the PRD in Southern China and is influenced by

anthropogenic emissions generated locally as well as transported from the other parts of the PRD and the northern China under the sub-tropical monsoon influence. PM pollution was observed to have a clear seasonal pattern, with lower concentrations in summer and higher in fall and winter. In summer, the southern oceanic air mass dominates, bringing cleaner air mass, and local sources play an important role, while in fall and winter, northerly winds prevail, bringing pollutants from the inland areas to HK. An increasing trend in atmospheric oxidation capacity has been observed in this region, signified by the increasing

trend of the surface ozone concentration from 2006-2019 (Li et al., 2022).

Previous continuous measurements in HK and elsewhere are mainly based on the high time resolution online mass spectrometry such as aerosol mass spectrometer (AMS) or aerosol chemical species monitor (ACSM). In these studies, the bulk organic aerosol (OA) was quantified and further separated into sub-groups of OA according to degree of oxidation and broad source origins, such as primary organic aerosol-POA (e.g., hydrocarbon-like organic aerosol-HOA, cooking organic

aerosol-COA and biomass burning organic aerosol-BBOA) and oxygenated organic aerosol-OOA (e.g., semi-volatile oxygenated organic aerosol-SVOOA, and low-volatility oxygenated organic aerosol-LVOOA). For example, Lee et al. (2015) and Sun et al. (2016) examined the chemical characteristics of $PM_1$ at a roadside station in HK in 2013 and pointed out the importance of COA and HOA contributions to the PM level at urban roadside environment. Li et al. (2015) examined the seasonal characteristics of $PM_1$ at a suburban site in HK from April 2011 to March 2012 and found that annual chemical

composition of $PM_1$ in HK were dominated by organic matter (OM) and sulfate and the background OM was mainly from secondary origins. Their study also emphasized the influence of air mass origins in the seasonal characteristics of the PM composition. Furthermore, Qin et al. (2016) examined evolution of PM characteristics during episodes across the seasons during four 1-month campaigns in 2011 at the same site, and identified three types of episodes: liquid water content episodes, high solar irradiance episodes, and long-range transport episodes. Their study revealed that both regional transport and

secondary formation contributed to high PM levels during the episodes at this site. However, the specific molecular information



was not available from those studies relying on AMS or ACSM data, making it difficult to extract precursor-specific information.

Different from AMS or ACSM, the Thermal desorption Aerosol Gas chromatography-mass spectrometer (TAG) continuously monitors individual organic compounds in ambient aerosol, including SOA products derived from specific volatile organic
compound (VOC) precursors. The precursor-specific SOA tracers can provide valuable molecule-level insight into the SOA formation processes and source origins. Important VOC precursors include major biogenic VOCs (i.e., isoprene, monoterpene and sesquiterpene) and anthropogenic aromatics such as benzene and toluene. Recent studies also suggested that phenol or substituted phenol, intensively emitted during biomass burning (BB), can produce 4-nitrocatechol resulting from reactions with OH under moderate $NO_x$. Thus, 4-nitrocatechol can serve to indicate BB-derived SOA formation (Finewax et al., 2018).
The formation mechanism of these VOC-specific SOA tracers has been well documented in the smog chamber studies (e.g., Claeys et al., 2004; Jaoui et al., 2007). For example, isoprene, the most abundant biogenic VOC on the global scale, forms various products via different formation pathways and the product distribution varies with atmospheric conditions. Under low-$NO_x$ conditions isoprene reacts with OH and $HO_2$ radicals and produces isoprene epoxydiols, subsequently leading to the formation of 2-methyltetrols and $C_5$-alkene triols in the aerosol. Under high-$NO_x$ conditions isoprene reacts with OH radical
and $NO_x$, producing methacrylic acid epoxide and subsequently forming the high-$NO_x$ product 2-methylglyceric acid in the aerosol. Monoterpene oxidation products contain both the early generation products (e.g., pinic acid) and later generation products (e.g., 3-methyl-1,2,3-butanetricarboxylic acid; 3-MBTCA) (e.g., Szmigielski et al., 2007). The ratio of pinic acid to 3-MBTCA can be used to indicate the aging of monoterpene SOA (Ding et al., 2012).

Quantification of the SOA tracers under field conditions have been so far carried out mainly through the offline filter samples
and subsequent laboratory Gas Chromatography-Mass Spectrometry (GC-MS) or Liquid Chromatography-Mass Spectrometry (LC-MS) measurements (e.g., Chow et al., 2016; Hu et al., 2008), due to the limited availability of the TAG systems. The field studies relying on filter-based tracer measurements mainly focus on variation characteristics of the SOA products and the influential factors of SOA formation on the seasonal temporal scale and on comparing spatial variations (Ding et al., 2012, 2016; Hu et al., 2008). The inherent low time resolution of the off-line filter measurements hinders the understanding of the
chemistry and formation mechanism at the hourly temporal scale. In this study, we report a series of SOA tracers measured by TAG at a bihourly resolution at a suburban site in HK from the end of Aug. to Dec. 2020. The objective is to investigate the chemical evolution of SOA tracers originated from biomass burning, anthropogenic and biogenic emissions under city-wide high $PM_{2.5}$ episodes across different seasons. Results from this study will help refine control strategies for future air quality improvement in HK and shed insights into precursor-SOA product dynamics for atmospheric environments under mixed urban
and regional pollution similar to our study location.





## 2 Sampling and measurement

The HKUST supersite is located on the campus of HKUST, on the hillside of the Clear Water Bay on the east coast of Sai Kung in New Territories, HK (22.33°N, 114.27°E). The site is in a low-density residential neighborhood. The nearby commercial and urban centers are 5-10 km away. There are little local emissions around, except a construction site and a small
canteen in the vicinity. During the study period, due to the pandemic, the cooking activities have been reduced to a minimum. The construction activities didn't emit the organic-rich particles and didn't influence the organic measurement. Thus, the sampling site can be considered as a background site in HK.

In this study, we focus on the comprehensive chemical speciation measurement conducted at HKUST supersite from 30 Aug. to 31 Dec. 2020. Bihourly organic molecular markers were measured at every even hour by the Aerodyne TAG standard alone
system coupled to an Agilent GC-MS (GC 7890B-MS 5977B). Detailed description about the instrument can be found in our previous work (He et al., 2020a; Wang et al., 2020a). TAG is capable of measuring over 100 different semi-volatile organic species including both nonpolar (e.g., alkanes, polyaromatic hydrocarbons (PAHs), and hopanes) and polar species (e.g., saccharides, aromatic acids, and carboxylic acids) in the aerosol phase. Among these individual TAG-measured organics, here we select to examine the abundant VOC-specific SOA tracers including six isoprene SOA tracers, six monoterpene SOA
tracers, one sesquiterpene SOA tracer, one monoaromatic SOA tracer, one naphthalene/methylnaphthalene SOA tracer, one BB-derived SOA tracer and one BB-sourced POA tracer (i.e., levoglucosan).

In addition to the TAG-measured individual organic compounds, other $PM_{2.5}$-related measurements include hourly $PM_{2.5}$ mass concentrations by a Sharp Monitor (Model 5030i; Thermo Fisher Scientific), major inorganic ions (sulfate, nitrate, ammonium, and chloride) by Monitor for AeRosols and Gases in ambient Air (MARGA 1S; Metrohm AG), carbonaceous components
(OC and EC) by a semi-continuous OC/EC analyzer (model RT-3179; Sunset Laboratory Inc.), and elemental species by an online X-Ray Florescence spectrometer (Xact 625i Ambient Continuous Multi-metals Monitor; Cooper Environmental Services). The biogenic and aromatic VOC precursors including isoprene, α-pinene, and BTEX (benzene, toluene, ethylbenzene, and xylene) were sampled by stainless canisters and analyzed by GC-FID/ECD/MSD system, with sample collection occurring at 9:00, 12:00 and 15:00 every day. Gas pollutants including $O_3$, $SO_2$ and $NO_x$ were measured by the gas
analyzers. Meteorological parameters including solar radiance, RH, temperature, precipitation, mixing height and wind data were measured by the 10-m AWS tower at the sampling site.

The air quality monitoring network in HK was operated by the Hong Kong Environment Protection Department (HKEPD). The city-wide network consists of 15 general stations and 3 roadside stations. Among the 15 general stations, 10 are in New Territories (i.e., NH, ST, TP, YL, TM, TC, TW, KC, TK, MB), 2 in Kowloon (i.e., SP and KT), and 3 in Hong Kong Island
(i.e., CW, EN and SN). One station (MB-Tap Mun) is a rural background site located on the isolated grass island in the northeastern HK, and others are either new town or urban stations with different microenvironments. Detailed description about the individual site characteristics is shown in Text S1. Hourly $PM_{2.5}$ mass concentrations from the 15 stations were retrieved from the HKUST supersite database (http://envf.ust.hk/dataview/metplot/current/index.py) and used for this work.



Hourly $PM_{2.5}$ mass concentrations and RH data at the HKUST supersite were biased during the studied time period. We did

the correction, and the details are shown in Text S2. $PM_{2.5}$ and gas pollutant data at the HKUST supersite were only available

after Sep. 2020, and the data at a similar rural station-MB (Tap Mun, 10 km to HKUST supersite) were

used as surrogate for days before Sep. 2020 (Figure S5).

## 3 Results and discussion

### 3.1 Classification of PM episodes and its spatial variability

Generally, a city-wide high PM episode is of more public concern than pollution at a single station. To examine PM pollution

in the whole city, we evaluated the hourly $PM_{2.5}$ data across the 15 general air quality monitoring stations of HKEPD. The

HKUST observation period starts from 30 Aug. to 31 Dec. 2020, thus we examined air quality data at the 15 HKEPD stations

from 10 Jul. to 31 Dec. 2020. The upper-level wind direction, together with sea level pressure and dew point date, is used to

determine the season division dates (Yu, 2002), and the details are described in Text S3. Specifically, the study period spanned

three seasons, i.e., summer (10 Jul. -7 Oct.), fall (8 Oct.-28 Nov.), and winter (29 Nov.-31 Dec.). Generally, $PM_{2.5}$ concentration

varied synchronously among different sites, regardless of urban or background sites, with correlation coefficient ($R_p$) ranging

0.71-0.92 during the non-episodic period, suggesting the regional characteristics of PM pollution in HK (Figure S3).

In this work, we define that a $PM_{2.5}$ pollution episode occurred when the $PM_{2.5}$ concentration was higher than 35 µg/m³ (24-

hour standard) for at least consecutive six hours at three or more stations. By this screening criterion, eleven PM episodes were

identified in this study, with one occurred in summer (i.e., EP1), five in fall (i.e., EP2-6) and five in winter (i.e., EP7-11). Table

1 lists the statistical summary of the episodic and non-episodic $PM_{2.5}$ averages, meteorological conditions, and gas pollutants

$O_3$ and $NO_x$ during the examined period (i.e., 10 Jul. to 31 Dec. 2020). Among the identified episodes, five are short episodes

lasting less than one day with three of them mainly occurring in the nighttime (denoted as EP6N, EP7N, and EP11N), and the

rest last much longer, ranging from one day to one week. The highest $PM_{2.5}$ pollution was observed under EP1 in summer, and

the city-wide average concentration was 37.6 µg/m³ during this episode. The lowest occurred under EP9 in winter (28.4 µg/m³).

The PM level during episodes was more than two times higher than that of non-episodic average (12.5 µg/m³). Wind speeds

were generally higher than 2 m/s during the episodes, except for EP1 and EP6N. High concentration of $O_3$ was observed under

EP1-6 and EP10, indicating higher atmospheric oxidation capacity. EP11N showed high concentrations of $NO_x$, which may

be attributed to the enhanced local vehicle emissions.

Figure 1 shows the spatial variation of average $PM_{2.5}$ during individual episodes and the remaining non-episodic hours. The

max-to-min ratio in Table 1 shows the difference of PM level across the 16 stations by normalizing the maximum average PM

concentration against the minimum PM average among the 16 stations for each episode. A ratio close to 1 suggests the

uniformity of PM pollution across the whole city while a larger ratio indicates the spatial gradient of PM pollution. A ratio of

1.6 was observed for non-episodic periods, with a spatial pattern of northwestern stations (e.g., TM, YL and TW) > central

stations (e.g., KT, TP and NH) > eastern/southeastern stations (e.g., TK and SN) (Figure 1). This pattern suggests the consistent



influence of regional transport from northern PRD region to HK. Episodes during summer to early fall (i.e., EP1-4) showed a slightly lower max-to-min ratio (~1.4), suggesting city-wide pollution characteristics. While for most winter episodes, an enhanced ratio was observed (1.9-4.3), signifying a larger concentration gradient in winter. The spatial variation during the winter episodes generally followed the trend of non-episodic period, with northwestern stations showing higher PM level than

the eastern/southeastern stations and that only a few exceptions were noted. For example, EP7N showed the highest PM level at three northern stations (TM, MB and TP), with concurrent high wind speed observed (6.13 m/s), signaling regional input from northern inland. EP10 showed the largest spatial difference (i.e., the max-to-min ratio of 4.3). During EP10, enhanced PM level was observed at the three northwestern stations (TM, YL and TC) while $PM_{2.5}$ at other stations didn't show obvious increase. Such a spatial pattern suggested influence of some local pollutions around this district. In contrast, in EP11N higher

concentrations were recorded at stations in the urban center of HK (i.e., CW, EN, KC, SP, KT, and TK) than the northwestern stations, which may be attributed to the enhanced local vehicular emissions during the end of the year holidays (i.e., 29-30 Dec.). The above results suggest that during summer to early fall, the PM episodes showed a more homogeneity feature, while in winter spatial heterogeneity of the episodes was more notable. The latter implies that air quality monitoring at a single station cannot represent the pollution status for the entire city.

**Table 1. Statistical summary of $PM_{2.5}$ at 15 HKEPD air quality monitoring stations and the HKUST supersite during the 11 episodes and the remaining non-episodic hours during the period of 10 Jul.-31 Dec. 2020. Meteorological parameters and gas pollutant data are from HKUST supersite.**

| Episodes | Season | Time period | Duration (h) | WS (m/s) | T (oC) | RH (%) | O₃ (ppb) | NOₓ (ppb) | Avg. $PM_{2.5}$, µg/m³ | | | |
|---|---|---|---|---|---|---|---|---|---|---|---|---|
| | | | | | | | | | City-wide avg. | Min. | Max. | Max-to-Min ratio |
| EP1 | Summer | Sep-01 12:00 PM - Sep-04 3:00 PM | 76 | 1.33 | 29.7 | 78.0 | 63.4 | / | 37.6 | 32.1 | 44.2 | 1.4 |
| EP2 | Fall | Oct-30 7:00 AM - 7:00 PM | 13 | 2.72 | 23.2 | 79.0 | 48.5 | 12.6 | 32.6 | 26.1 | 39.9 | 1.5 |
| EP3 | Fall | Nov-02 7:00 AM - Nov.-04 9:00 PM | 63 | 3.93 | 23.2 | 65.5 | 57.6 | 12.0 | 30.7 | 26.6 | 37.8 | 1.4 |
| EP4 | Fall | Nov-06 11:00 AM - Nov-10 10:00 PM | 108 | 3.36 | 24.2 | 56.9 | 69.0 | 13.1 | 33.0 | 28.4 | 41.0 | 1.4 |
| EP5 | Fall | Nov-24 12:00 PM - 7:00 PM | 8 | 3.44 | 22.4 | 79.1 | 58.4 | 8.22 | 29.7 | 22.1 | 43.0 | 1.9 |
| EP6N | Fall | Nov-26 4:00 PM - Nov-27 1:00 AM | 10 | 1.36 | 20.8 | 89.6 | 58.5 | 7.20 | 29.9 | 17.2 | 44.9 | 2.6 |
| EP7N | Winter | Dec-03 1:00 AM - 10:00 AM | 10 | 6.13 | 15.6 | 67.7 | 26.2 | 11.2 | 31.0 | 24.3 | 38.9 | 1.6 |
| EP8 | Winter | Dec-05 2:00 AM - Dec-13 0:00 AM | 191 | 2.59 | 18.7 | 71.9 | 41.8 | 13.9 | 33.2 | 25.6 | 47.9 | 1.9 |
| EP9 | Winter | Dec-19 1:00 PM - Dec-25 10:00 PM | 154 | 3.62 | 16.3 | 68.4 | 38.2 | 11.3 | 28.4 | 20.6 | 41.9 | 2.0 |
| EP10 | Winter | Dec-27 11:00 AM - Dec-28 11:00 AM | 25 | 1.89 | 20.5 | 58.0 | 61.7 | 6.69 | 28.9 | 15.7 | 66.7 | 4.3 |
| EP11N | Winter | Dec-29 8:00 PM - Dec-30 4:00 AM | 9 | 3.70 | 18.5 | 72.1 | 32.2 | 21.3 | 33.8 | 22.8 | 43.4 | 1.9 |
| non-EP | / | Jul-10 00:00 AM - Dec-31 11:00 PM | 3533 | 2.88 | 25.0 | 78.9 | 44.8 | 9.10 | 12.5 | 10.4 | 16.9 | 1.6 |





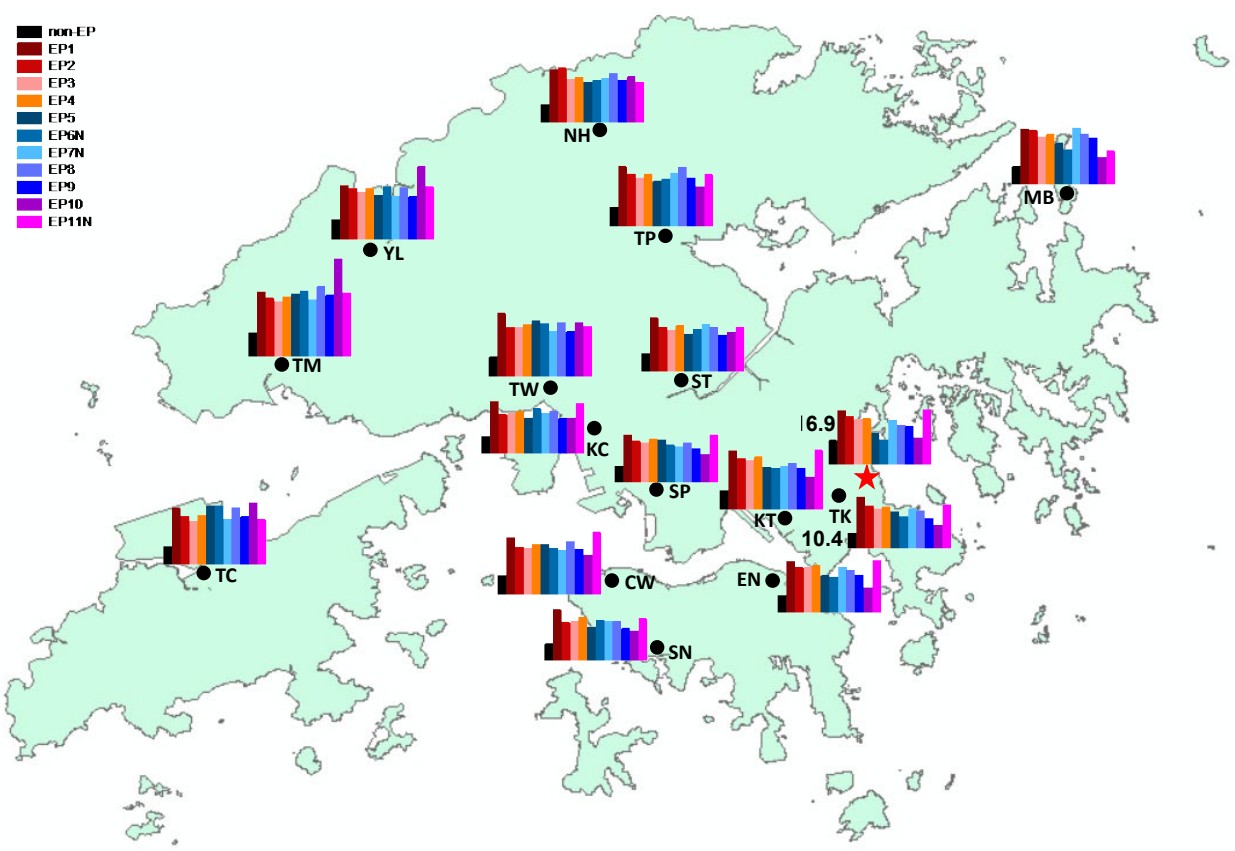

**Figure 1. Geographical location of the 15 HKEPD general air quality monitoring stations (black dot) and the HKUST supersite (red star) in HK. Among the 15 HKEPD stations, MB is the rural site while others are general urban stations with different microenvironments. Column plot shows the average PM$_{2.5}$ concentrations (μg/m³) during non-episodic hours (in black, with numbers indicating the concentration values) and the eleven episodes (in various colors).**

**3.2 Overview of PM speciation measurement at the HKUST supersite**

Compared with the total PM mass, the pollution characteristics of individual components, especially organics, may be quite different at different locations. The full PM composition data are only available at HKUST supersite starting from 30 Aug. to 31 Dec. 2020. Figure 2 shows the time series of PM$_{2.5}$ and its major chemical components, select individual SOA tracers and its VOC precursors. The source-specific organic markers include (1) BB-derived POA tracer (levoglucosan) and SOA tracer (4-nitrocatechol), (2) anthropogenic SOA tracers including naphthalene/methylnaphthalene SOA tracer (phthalic acid) and monoaromatic SOA tracer, and (3) biogenic SOA tracers including six isoprene SOA tracers, six α-pinene SOA tracers and one β-caryophyllene SOA tracer. The SOA tracers generally showed higher concentrations during the daytime (Figure S8), in consistence with their secondary origins. Moderate to good correlation was observed among the SOA tracers and with sulfate, suggesting the regional feature (Figure S9).





To characterize chemical features in the formation of PM episodes, we examined the PM composition before and during each

episode. The selection of the before-period for each episode is shown in Table S2 and Figure S10. We select the before-period time windows primarily based on the principle that the before-period duration was comparable with that for the episode and that some time interval (> 12 h) was given between the before-period and the immediately past episode for avoiding residual influences from the previous episode. Air mass origins for each time window pair of episode vs. before-period were shown in Figure S11. Similar air mass origins from northern continent were observed during the before and episodic period for most

episodes (except EP1), excluding sudden change of air mass origins as the leading cause for the rapid increase of PM level. Mass increment ratio (MIR), calculated as the mass concentration during the episode divided by that before the episode, was used to evaluate the concentration change of different chemical species during the episode. A MIR>1 suggests the increase of the concentration during the episode, and vice versa.

Figure S12 shows the average PM composition for the before-period and during episodes. EP6N and EP10 are not obvious at

this site, with average PM <20 μg/m$^3$ and only slightly higher than the corresponding before-period. The PM composition showed sulfate and OM were the major components throughout the measurement period while increased nitrate and ammonium concentrations were recorded in winter due to the lower temperatures facilitating partitioning of ammonium nitrate to particle phase. Generally, the MIR values are higher for episodes occurring in summer and early fall (EP1-4) while less differences in the PM mass were observed before and during episodes for the winter episodes (except EP9), due to the higher background

PM level. For the major PM constituents, MIR>1 was generally observed during all episodes except for EP6N&10. Nitrate showed the largest MIR values, especially in EP1 and EP11N.

Increased formation of secondary inorganic aerosol during PM$_{2.5}$ episodes have been extensively reported in the literature (e.g., Liu et al., 2020; Yun et al., 2018), and thus will not be our focus. In the later sections, we will examine in detail the chemical evolution of the precursor-specific SOA tracers during the episodes.



**Figure 2.** Time series of PM$_{2.5}$, its major components, select POA and SOA tracers and its VOC precursors during the observation period (30 Aug. – 31 Dec. 2020) at the HKUST supersite. The episodic periods are shaded in grey, and the before-episode periods are colored in yellow. The full name of the SOA tracers are shown in Table 2.



**Table 2. Summary of average concentration of PM$_{2.5}$, its major components, select POA and SOA tracers and its VOC precursors measured at HKUST supersite under each episode and different seasons during 30 Aug. – 31 Dec. 2020.**

| | EP1 | EP2 | EP3 | EP4 | EP5 | EP6 | EP7 | EP8 | EP9 | EP10 | EP11 | Summer | Fall | Winter |
|---|---|---|---|---|---|---|---|---|---|---|---|---|---|---|
| **Major component (µg/m³)** | | | | | | | | | | | | | | |
| PM$_{2.5}$ | 37.5 | 33.0 | 31.0 | 31.6 | 22.1 | 17.2 | 30.5 | 27.2 | 26.3 | 18.7 | 37.8 | 13.0±10.3 | 19.2±8.7 | 21.9±9.1 |
| OM (1.4×OC) | 11.7 | / | / | / | 6.46 | 5.67 | 5.22 | 8.04 | 7.46 | 8.07 | 9.47 | 4.65±2.98 | 5.52±1.90 | 6.58±2.66 |
| EC | 1.87 | / | / | / | 0.61 | 0.72 | 0.96 | 1.38 | 1.50 | 0.78 | 1.71 | 0.78±0.62 | 0.85±0.33 | 1.18±0.79 |
| NO$_3^-$ | 4.28 | 2.09 | / | 0.16 | / | 1.93 | 4.01 | 4.28 | 5.38 | 2.05 | 7.74 | 1.12±1.39 | 1.04±0.81 | 3.92±2.48 |
| SO$_4^{2-}$ | 11.4 | 8.60 | 7.60 | 6.91 | / | 4.30 | 8.77 | 6.67 | 5.39 | 3.26 | 4.92 | 4.31±3.23 | 4.51±2.51 | 5.25±2.72 |
| NH$_4^+$ | 5.21 | 1.42 | / | / | / | / | 4.76 | 3.66 | 3.26 | 1.23 | 3.67 | 1.29±1.59 | 0.51±0.65 | 2.84±1.59 |
| **POA and SOA tracers (ng/m³)** | | | | | | | | | | | | | | |
| Levoglucosan | 35.5 | / | 35.1 | 26.1 | 30.5 | 23.2 | 28.6 | 35.9 | 35.5 | 24.7 | 31.4 | 10.6±12.8 | 25.4±17.3 | 29.6±16.7 |
| 4-nitrocatechol | 6.60 | / | 7.12 | 6.39 | 4.92 | 3.55 | 2.26 | 8.10 | 6.56 | 3.44 | 7.50 | 1.36±3.45 | 3.67±4.12 | 6.12±6.80 |
| Phthalic acid | 33.8 | / | 28.0 | 24.7 | 31.0 | 21.0 | 18.9 | 20.4 | 17.3 | 8.59 | 14.4 | 12.4±18.1 | 16.8±12.5 | 15.3±11.0 |
| 2,3-dihydroxy-4-oxopentanoic acid (DHOPA) [a] | 2.84 | / | 3.01 | 1.92 | 1.33 | 1.94 | 0.32 | 0.36 | 0.24 | 0.10 | 0.17 | 1.01±1.50 | 1.17±1.09 | 0.26±0.22 |
| **α-pinene SOA tracers** | | | | | | | | | | | | | | |
| Pinic acid [b] | 19.1 | / | 9.92 | 8.66 | 10.5 | 12.6 | 1.82 | 4.55 | 3.92 | 3.31 | 6.28 | 8.06±9.02 | 6.42±4.55 | 3.26±2.66 |
| Pinonic acid | 6.49 | / | 2.93 | 6.65 | 1.53 | 1.29 | 0.77 | 1.35 | 1.24 | 0.53 | 0.56 | 1.40±2.34 | 2.10±2.33 | 0.83±0.95 |
| 3-hydroxyglutaric acid (3-HGA) [b] | 9.45 | / | 7.19 | 6.10 | 5.70 | 6.65 | 0.94 | 1.85 | 1.29 | 0.94 | 2.12 | 4.52±5.12 | 3.64±2.78 | 1.17±1.01 |
| 3-acetylglutaric acid (3-AGA) [b] | 10.4 | / | 3.61 | 4.26 | 3.22 | 2.53 | 0.80 | 1.42 | 1.11 | 0.38 | 2.21 | 2.44±3.39 | 2.23±1.84 | 0.90±0.88 |
| 3-hydroxy-4,4-dimethyl glutaric acid (3-HDGA) [b] | 27.3 | / | 12.4 | 11.4 | 4.42 | 6.34 | 0.55 | 1.04 | 0.94 | 0.67 | 1.56 | 10.1±13.4 | 5.80±4.40 | 0.81±0.72 |
| 3-methyl-1,2,3-butanetricarboxylic acid (3-MBTCA) [b] | 19.3 | / | 13.8 | 16.6 | 10.9 | 7.29 | 1.39 | 2.80 | 1.90 | 0.63 | 1.35 | 5.64±7.05 | 8.00±6.63 | 1.65±1.51 |
| Σα-pinene SOA tracers | 92.0 | / | 49.8 | 53.7 | 36.3 | 36.7 | 6.27 | 13.0 | 10.4 | 6.46 | 14.1 | 32.2±36.8 | 28.2±19.7 | 8.62±6.86 |
| **isoprene SOA tracers** | | | | | | | | | | | | | | |
| 2-methylglyceric acid (2-MGA) [b] | 4.22 | / | 1.73 | 2.68 | 0.64 | 0.73 | 0.10 | 0.28 | 0.25 | 0.10 | 0.33 | 0.55±1.27 | 0.63±1.00 | 0.20±0.28 |
| 2-methylthreitol (2-MT) [c] | 13.0 | / | 0.58 | 0.58 | 0.37 | 0.35 | 0.41 | 0.25 | 0.14 | 0.05 | 0.10 | 1.62±4.42 | 0.53±0.47 | 0.14±0.13 |
| 2-methylerythritol (2-MET) [c] | 38.3 | / | 1.96 | 1.89 | 1.53 | 1.56 | 1.21 | 0.95 | 0.66 | 0.21 | 0.45 | 5.31±14.1 | 2.02±1.88 | 0.57±0.49 |
| cis-2-methyl-1,3,4-trihydroxy-1-butene (cis 2-MBT) [b] | 10.1 | / | 0.55 | 1.35 | 0.34 | 1.08 | 0.09 | 0.15 | 0.14 | 0.03 | 0.37 | 0.84±3.12 | 0.39±0.58 | 0.14±0.16 |
| 3-methyl-2,3,4-trihydroxy-1-butene (3-MBT) [b] | 18.9 | / | 0.74 | 1.95 | 0.63 | 0.52 | 0.10 | 0.25 | 0.20 | 0.06 | 0.68 | 1.61 5.80 | 0.51 0.80 | 0.13 0.23 |
| trans-2-methyl-1,3,4-trihydroxy-1-butene (trans 2-MBT) [b] | 7.17 | / | 2.45 | 4.45 | 1.93 | 2.11 | 0.44 | 0.96 | 0.89 | 0.31 | 1.54 | 1.12±2.33 | 1.32±1.62 | 0.61±0.65 |
| Σisoprene SOA tracers | 91.7 | / | 8.01 | 12.9 | 5.44 | 6.35 | 2.34 | 2.84 | 2.28 | 0.74 | 3.47 | 11.1±27.2 | 5.40±5.29 | 1.80±1.57 |
| β-caryophyllinic acid [b] | 5.95 | / | 4.17 | 3.21 | 3.65 | 2.55 | 1.96 | 1.56 | 1.49 | 0.23 | 0.99 | 2.60±3.55 | 2.57±1.91 | 1.20±0.92 |
| **VOC precursors (ppb)** | | | | | | | | | | | | | | |
| BTEX (sum of benzene, toluene, ethylbenzene, and xylene) | 1.16 | 1.69 | 1.54 | 1.11 | 0.80 | 0.47 | 1.21 | 1.50 | 1.16 | 0.72 | / | 0.68±0.86 | 0.97±0.58 | 1.41±0.68 |
| isoprene | 1.83 | 0.22 | 0.32 | 0.47 | 0.29 | 0.069 | 0.013 | 0.082 | 0.051 | 0.20 | / | 0.93±0.95 | 0.34±0.32 | 0.08±0.07 |
| α-pinene | 0.036 | 0.018 | 0.011 | 0.010 | 0.004 | 0.000 | 0.005 | 0.007 | 0.003 | 0.005 | / | 0.03±0.04 | 0.02±0.02 | 0.005±0.004 |

[a] Quantified using azelaic acid. [b] quantified using pinonic acid. [c] quantified using levoglucosan as surrogate.





### 3.3 Biomass burning POA and SOA tracers

Levoglucosan, originating from the pyrolysis of cellulose and hemicellulose, has been widely used as a BB POA tracer
(Simoneit et al., 1999). In our dataset, a moderate correlation of 4-nitrocatechol with levoglucosan was observed ($R_p$: 0.43), in line with their common material origin from BB. It is also noted that 4-nitrocatechol was moderately correlated with benzene ($R_p$: 0.52) and toluene ($R_p$: 0.50), implicating these anthropogenic VOCs as notable contributing precursors to 4-nitrocatechol as well. Furthermore, 4-nitrocatechol and $NO_x$ were moderately correlated ($R_p$: 0.48), suggesting the importance of $NO_x$ oxidation of the aromatic VOC precursors.

Seasonal variation of 4-nitrocatechol showed the highest concentration in winter (6.12±6.80 ng/m$^3$), followed by fall (3.67±4.12 ng/m$^3$) and summer (1.36±3.45 ng/m$^3$). Levoglucosan showed comparable high concentrations in winter and fall (29.6±16.7 and 25.4±17.3 ng/m$^3$, respectively), which were more than two times higher than that in summer (10.6±12.8 ng/m$^3$). It is noted that the levoglucosan concentration in this study was lower than previous offline filter-based measurements from fall to winter in 2010-2012 in HK (avg. 96.8 ng/m$^3$) while the concentration of 4-nitrocatechol was comparable (avg. 3.42
ng/m$^3$ in Chow et al. 2016). This likely reflects that 4-nitrocatechol has precursor sources other than BB and joint measurements of potential precursors (e.g., catechol, phenol, benzene) in the future would help to discern the relative importance of precursors from BB versus anthropogenic sources.

Figure 3a compares the average concentration of levoglucosan and 4-nitrocatechol for each pair of before-period and episodic period. The variation of 4-nitrocatechol was generally in sync with the variation of $NO_x$ between the before-period and episodic
period, signifying the importance of $NO_x$ influence. Levoglucosan and 4-nitrocatechol jointly showed higher concentration under most episodes except EP5 and EP6N, indicating BB as a frequent important contributor to episodic increase in PM$_{2.5}$. Figure 3b shows the MIR distribution of levoglucosan and 4-nitrocatechol during each episode. MIR>1 was observed for both levogucosan and 4-nitrocatechol during summer and early fall episodes (EP1-4), with the latter showing noticeably larger MIR values (3-7). The observations suggested enhanced contributions from both primary BB emissions and BB-derived SOA,
especially secondary formation during EP1-4. For EP5 and EP6N, the concentration of levoglucosan was higher in the before-period compared with the episode hours. During winter episodes (EP7-11), MIR values were lower (<3), which means lesser differences between the before-period and episodic hours. In the early winter episodes (EP7-9), the MIR values of levoglucosan were higher than 4-nitrocatechol, suggesting more important contribution from primary BB emissions during these episodes. In EP11N, a higher MIR value was observed for 4-nitrocatechol than levoglucosan, and the increase of 4-nitrocatechol was
accompanied by the increase of $NO_x$, suggesting enhanced nighttime secondary formation during this episode.

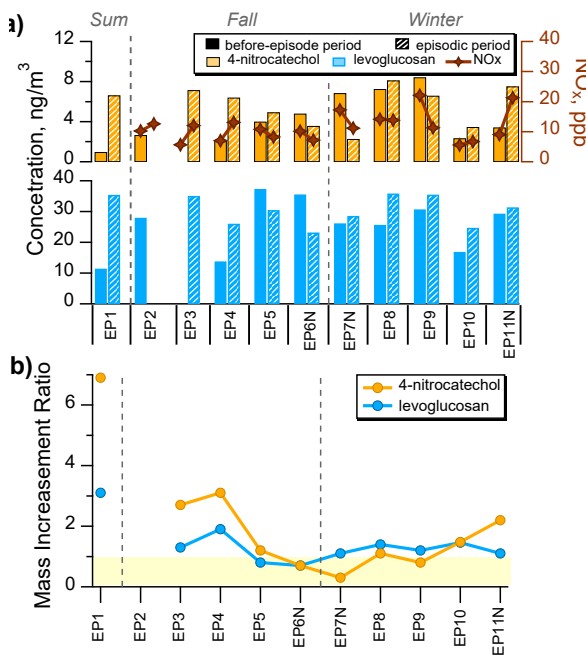

**Figure 3. (a) Comparison of average concentration of 4-nitrocatechol and levoglucosan during the before-episode periods (solid filling) and the episodic periods (pattern filling). (b) Mass increment ratios of 4-nitrocatechol and levoglucosan for each episode, with the light-yellow area marking the values less than 1.**

## 3.4 Anthropogenic SOA tracers

Two anthropogenic SOA tracers were measured in this study, i.e., phthalic acid and DHOPA. Phthalic acid is a SOA tracer from two-ring naphthalene/methylnaphthalene (Kleindienst et al., 2012), while DHOPA is a SOA tracer from monoaromatics such as BTEX (benzene, toluene, ethylbenzene, and xylene) (Al-Naiema and Stone, 2017). Previous studies showed that the main atmospheric loss of benzene and toluene is their photochemical reaction with OH radicals, with an atmospheric lifetime of 12.5 and 2.0 d, respectively (Prinn et al., 1987). Since there were few anthropogenic sources near the HKUST supersite, the measured BTEX should not be locally emitted but mainly transported from upwind mainland China due to their long atmospheric lifetime.

Phthalic acid showed slightly higher concentration in fall and winter ($16.8\pm12.5$ and $15.3\pm11.0$ ng/m$^3$) while lower concentration in summer ($12.4\pm18.1$ ng/m$^3$). In comparison, DHOPA showed a more notable seasonal variation, with the concentrations in fall and summer ($1.17\pm1.09$ and $1.01\pm1.50$ ng/m$^3$) being about 5 times that in winter ($0.26\pm0.22$ ng/m$^3$). DHOPA's VOC precursors (i.e., BTEX) showed the pattern of winter > fall > summer, with the seasonal average concentration at $1.41\pm0.68$, $0.97\pm0.58$ and $0.68\pm0.86$ ppb, respectively. Phthalic acid and DHOPA positively correlated with O$_3$ ($R_p$: 0.36 and 0.44), while DHOPA did not correlate with its aromatic VOC precursors ($R_p$: 0.03). The results suggest that oxidant level is a significant factor promoting the formation of both phthalic acid and DHOPA, consistent with their secondary origin.





Figure 4a shows the average concentration of phthalic acid and DHOPA quantified during the before-periods and episodic
       periods. Among the 11 episodes, higher concentration of the two SOA tracers were observed during summer and fall episodes
       (EP1-6), with episode-average values of 21.0-33.8 and 1.33-3.01 ng/m$^3$ for phthalic acid and DHOPA, respectively. During
       the winter episodes (EP7-11), the average concentrations of both SOA tracers were in a lower range, with DHOPA significantly
       lower (0.10-0.36 ng/m$^3$) and phthalic acid slightly lower (8.59-20.4 ng/m$^3$). Shown in Figure 4b, BTEX had an enhanced
presence during EP1-4 while no discernable elevation was detected during the remaining episodes in comparison with before-
       periods. DHOPA and phthalic acid also had higher MIR values in EP1-4 (MIR> 2) than the other episodes (MIR: 0.7-2), in
       line with the precursor-product relationship. The MIR values of DHOPA and phthalic acid exceeded unity in the remaining
       episodes, EP7N being an exception for DHOPA and EP11N being an exception for phthalic acid. This seeming discrepancy
       in concentration variation between precursor-product reflects that the key factors influencing formation of SOA tracers are not
limited to their VOC precursors.

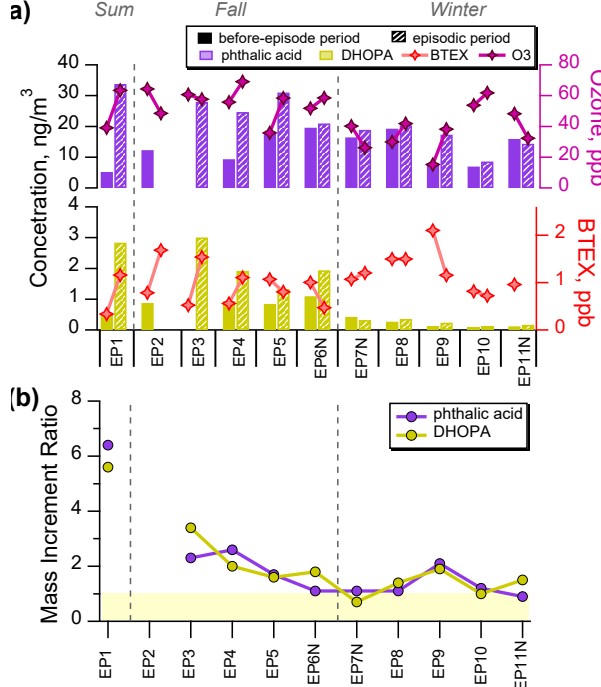

**Figure 4. (a) Comparison of average concentration of phthalic acid, DHOPA, BTEX, and ozone during the before-episode periods
(solid filling) and episodic periods (pattern filling). (b) Mass increment ratios of phthalic acid and DHOPA for each episodes, with
the light-yellow area marking the ratio values less than 1.**

**3.5 Biogenic SOA tracers**

       Three types of biogenic SOA tracers were quantified in this campaign, i.e., SOA tracers derived from isoprene, α-pinene, and
       β-caryophyllene. Previous studies estimated that the atmospheric lifetime of isoprene, α-pinene and β-caryophyllene was 1.3
       d, 4.6 h, and 2 min against ozone at a concentration of $7 \times 10^{11}$ cm$^{-3}$ (~30 ppb) and 1.4 h, 2.6 h, and 42 min against OH at 2 ×



$10^6$ cm$^{-3}$ (Atkinson and Arey, 2003), respectively. Thus, isoprene would preferentially react with OH, while β-Caryophyllene
will mainly react with O$_3$. Compared with the precursors, the chemical lifetime of the SOA tracers is much longer in the range
of 2-10 days (Nozière et al., 2015). This implies that the SOA tracers observed at the site can be significantly contributed by
regional/super-regional transport. The diurnal variation of the α-pinene and β-caryophyllene SOA tracers in this study showed
clearly enhanced concentrations during the daytime from 10:00-16:00 while the isoprene SOA tracers did not show a
discernable trend (Figure S8).

**3.5.1 Isoprene SOA tracers**

The sum concentration of isoprene SOA tracers was the highest in summer (11.1±27.2 ng/m$^3$), followed by fall (5.40±5.29
ng/m$^3$), and the lowest in winter (1.80±1.57 ng/m$^3$). Such a pattern was similar to the seasonality of isoprene ambient
concentration. At our site, isoprene was 0.93±0.95 ppb in the summer, 0.34±0.32 ppb in the fall, and 0.08±0.07 ppb in the
winter, in agreement with the temperature-dependent characteristic of isoprene emissions. Ambient isoprene is mainly
controlled by local emissions considering its short atmospheric lifetime. No correlation was observed between the isoprene
SOA tracers and isoprene or temperature, suggesting a significant part of the isoprene SOA tracers were likely brought to the
site via regional/super-regional transport. The isoprene SOA tracers were consistently higher during episodes than the before-
episode periods (Figure 5). During episodes, the highest concentration (91.7 ng/m$^3$) occurred in the summer episode (EP1), far
exceeding those in the fall episodes (average: 8.18 ng/m$^3$, range: 5.44-12.9 ng/m$^3$) and the winter episodes (average: 2.33
ng/m$^3$, range: 0.74-3.47 ng/m$^3$). The stark seasonal contrast was in line with the strong temperature-dependence of isoprene
emissions and consequent ambient concentrations.

A total of six major isoprene SOA tracers were measured in this work, namely 2-MGA, two 2-methyltetrols (2-MT, 2-MET),
and three C$_5$-alkene triols (*cis* 2-MTB, 3-MTB, and *trans* 2-MTB). Previous laboratory studies suggested that 2-MGA was
produced by NO$_x$-channel under high-NO$_x$ condition (hundreds of ppb) while 2-methyltetrols and C$_5$-alkene triols were formed
through HO$_2$-channel under low-NO$_x$ (several ppb level) or NO$_x$ free conditions (Claeys et al., 2004; Edney et al., 2005; Surratt
et al., 2010). 2-Methyltetrols could also be produced by isoprene ozonolysis in the presence of acidic aerosol (Riva et al., 2016)
and non-acidified sulfate aerosol (Kleindienst et al., 2007). During the whole sampling period, NO$_x$ was at a relatively low
level (10.1±7.52 ppb). Composition of isoprene SOA tracers consistently showed the dominance of C$_5$-alkene triols and 2-
methyltetrols (Figure 5a). This result suggested that the oxidation of isoprene with OH via HO$_2$-channel was dominant,
consistent with the NO$_x$ monitoring data.

The MIR values of isoprene SOA tracers, shown in Figure 5b, were generally the highest among all the SOA tracers measured
in all episodes except for EP7N, which occurred mainly in the nighttime. EP1-4 had much higher MIR values (4.1-11), clearly
indicating more enhanced isoprene SOA formation during the summer and early fall episodes. The winter episodes had lower
MIR values for the isoprene SOA tracers, suggesting much lower isoprene SOA formation likely as a result of the low
availability of the precursor.



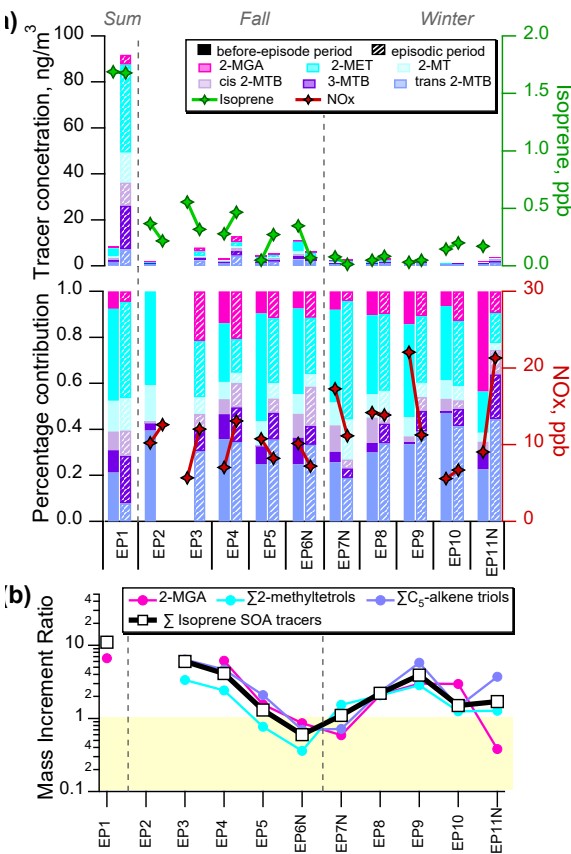

**Figure 5. (a)** Comparison of average concentration and molecular distributions of isoprene SOA tracers in the before-episode periods (solid filling) and the episodic periods (pattern filling). **(b)** Mass increment ratios of isoprene SOA tracers for each episode, with the light-yellow area marking the ratio values less than 1.

### 3.5.2 Monoterpene SOA tracers

Among the measured three types of biogenic SOA tracers, α-pinene SOA tracers had the highest abundance. The sum concentration of α-pinene SOA tracers was the highest in summer ($32.2\pm36.8$ ng/m$^3$), followed by fall ($28.2\pm19.7$ ng/m$^3$), then winter ($8.62\pm6.86$ ng/m$^3$). The temporal variation of α-pinene also showed higher abundance in summer ($0.03\pm0.04$ ppb) and fall ($0.02\pm0.02$ ppb), and much lower concentration in winter ($0.005\pm0.004$ ppb). Similar to the isoprene SOA tracers, we did not observe any correlations between the α-pinene SOA tracers and α-pinene, suggesting the α-pinene SOA tracers measured are not locally formed.

Episodic concentration of α-pinene SOA tracers showed the highest value during summer and fall episodes (EP1-6; average: 53.7 ng/m$^3$, range: 36.3-92.0 ng/m$^3$), more than 5 times higher than the winter episodes (EP7-11; average: 10.1 ng/m$^3$, range: 6.27-14.1 ng/m$^3$) (Figure 6a). A total of six α-pinene SOA tracers were quantified, including pinic acid, pinonic acid, 3-HGA, 3-HDGA, 3-AGA and, 3-MBTCA. Chamber studies showed that pinonic and pinic acid are the first generation products of α-pinene oxidation, while the other 4 tracers are of later generations (Szmigielski et al., 2007). The molecular distribution of the





α-pinene SOA tracers showed a clear seasonality. That is, 3-HDGA and 3-MBTCA were more abundant in summer and fall while pinic acid was the most abundant in winter. This agrees with the fact that the enhanced atmospheric oxidative capacity in summer and fall was conducive for more later generations of SOA products.

Similar to isoprene SOA tracers, α-pinene SOA tracers had MIR>1 in all episodes except for EP7N. EP1-4 and EP8-9 had much higher MIR values, suggesting the enhanced α-pinene SOA formation during the episodes (Figure 6b). In a previous offline filter-based study in HK, Hu et al. (2008) proposed that the formation of SOA was sensitive to the level of $O_3$ on the basis of observed positive correlations between secondary organic carbon and $O_3$. In this study, we also observed the positive correlation between α-pinene SOA tracers and $O_3$ ($R_p$: 0.26-0.45), supporting a significant role by the atmospheric oxidant in

the formation of monoterpene SOA.

   Pinic acid is an intermediate of α-pinene oxidation and can be further oxidized to 3-MBTCA (Claeys et al., 2007). The ratio of pinic acid/3-MBTCA (abbreviated as P/M hereafter) could be used to evaluate the aging processes of α-pinene SOA, with a lower P/M signaling more aged α-pinene SOA. A negative correlation was observed between 3-MBTCA and the P/M ratio, indicating that more high-generation products occur in more aged α-pinene SOA (Figure 6c). Figure 6d shows the temporal

variation of the P/M ratio for the before and during episodic period. The P/M ratio seasonality of fall ≈ summer < winter indicated more aged α-pinene SOA in fall and summer than in winter. Compared with the respective before--episode periods, EP1-5 and EP9 showed higher degree of aging of α-pinene SOA as indicted by their noticeably lower P/M ratios. In comparison with the P/M ratio obtained in other studies, the ratio in winter in this work (2.64 ± 1.91) is comparable to those measured in urban Shanghai (3.6 ± 1.5; He et al., 2020) and rural Guangzhou (3.02; Yuan et al., 2018) in winter, as well as that of fresh

SOA in chamber studies (1.51 - 3.21; Offenberg et al., 2007). The results suggest that the wintertime monoterpene SOA is generally relatively fresh.





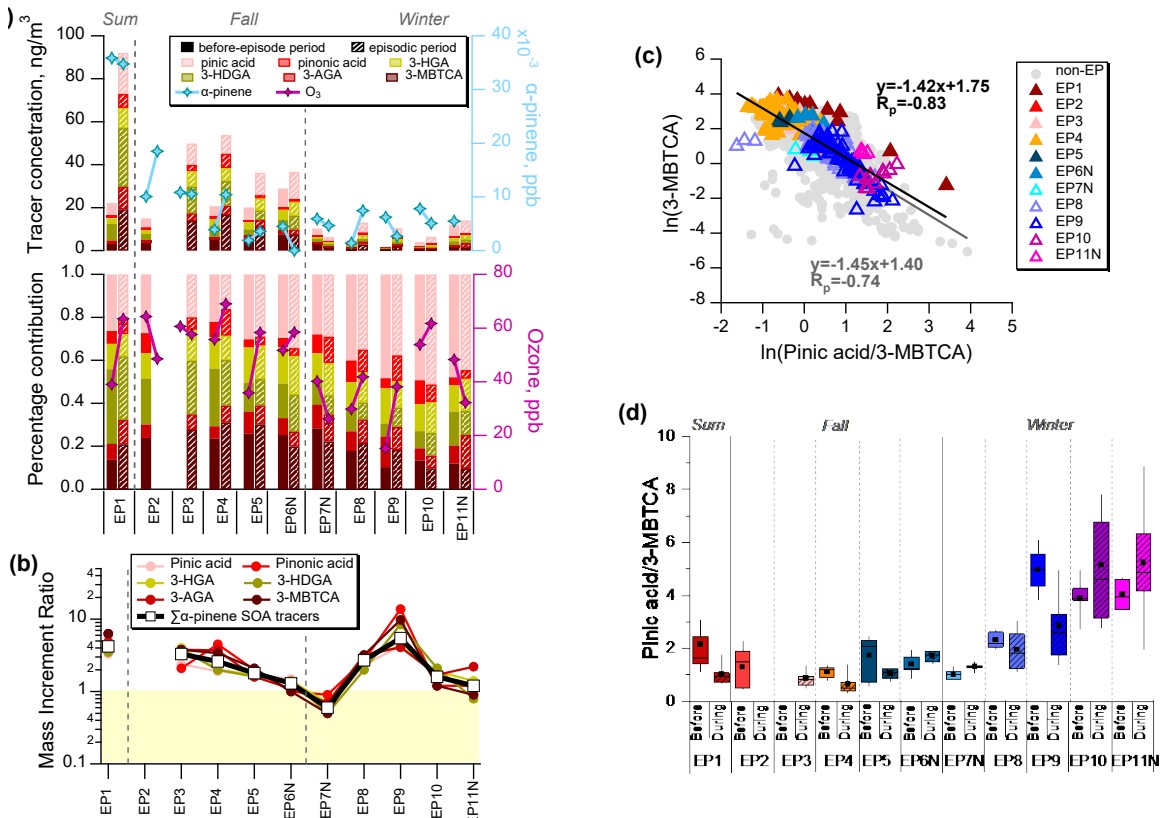

**Figure 6. (a) Comparison of average concentration and molecular distribution of α-pinene SOA tracers in the before-episode periods (solid filling) and the episodic periods (pattern filling). (b) Mass increment ratios of α-pinene SOA tracers for each episodes, with the light-yellow area marking the ratio values less than 1. (c) Correlation between ln(Pinic acid/3-MBTCA) and ln(3-MBTCA). (d) Distribution of the Pinic acid/3-MBTCA ratio during before-episode periods (solid filling) and the episodic periods (pattern filling). Squares and solid lines correspond to mean and median values, respectively, box indicates the 25th and 75th percentile, and whiskers are the 10th and 90th percentile.**

### 3.5.3 Sesquiterpene SOA tracers

Seasonal variation of β-caryophyllinic acid showed higher concentration in summer ($2.60 \pm 3.55$ ng/m$^3$) and fall ($2.57 \pm 1.91$ ng/m$^3$), which were ~2 times that in winter ($1.20 \pm 0.92$ ng/m$^3$). Due to the high reactivity towards ozone, the precursor β-caryophyllene is generally not detectable at the typical ambient ozone concentrations (~10-100 ppb). Its ambient concentration was unavailable in this work either. The episodic concentration of β-caryophyllinic acid showed higher value during summer and fall episodes (EP1-6; 2.55-5.95 ng/m$^3$) than those during the winter episodes (EP7-11; 0.23-1.96 ng/m$^3$) (Figure 7a). β-caryophyllinic acid also showed higher MIR values for summer and fall episodes (1.2-6), while for EP7-8&10, MIR<1 was observed (Figure 7b).

We observed that the concentration difference of β-caryophyllinic acid between the two sub-periods (i.e., EP1-6 vs EP7-11) was much less compared with those of α-pinene and isoprene SOA tracers. Previous chamber studies of the β-caryophyllene ozonolysis reaction suggested a number of first-generation products, such as aldehydes (e.g., β-caryophyllon aldehyde and β-





hydroxycaryophyllon aldehyde) and acids (e.g., β-caryophyllonic acid and β-caryophyllinic acid) (e.g., Chan et al., 2011). The first-generation ozonolysis products, which still contain a C=C double bond, can be oxidized to the second-generation products (e.g., β-nocaryophyllon aldehyde and β-hydroxynocaryophyllon aldehyde). In most field studies, the identification of other β-caryophyllene SOA tracers was rarely available, due to the lack of authentic chemical standards and the reference mass spectra. It is plausible that the higher concentration of β-caryophyllinic acid in both non-episodic and episodic periods in winter

compared with those in summer and fall could be a result of the lower atmospheric oxidative capacity in winter leading to less degradation of β-caryophyllinic acid. This speculation is supported by the good correlation between β-caryophyllinic acid and pinic acid ($R_p$: 0.81; Figure 7c). Figure 7c also suggests that sesquiterpene SOA in winter in HK is relatively fresh. For a more definitive tracking of aging degree of β-caryophyllene SOA, we recommend future efforts directed at laboratory characterization of its later generation products and joint field monitoring of multi-generation oxidation products.

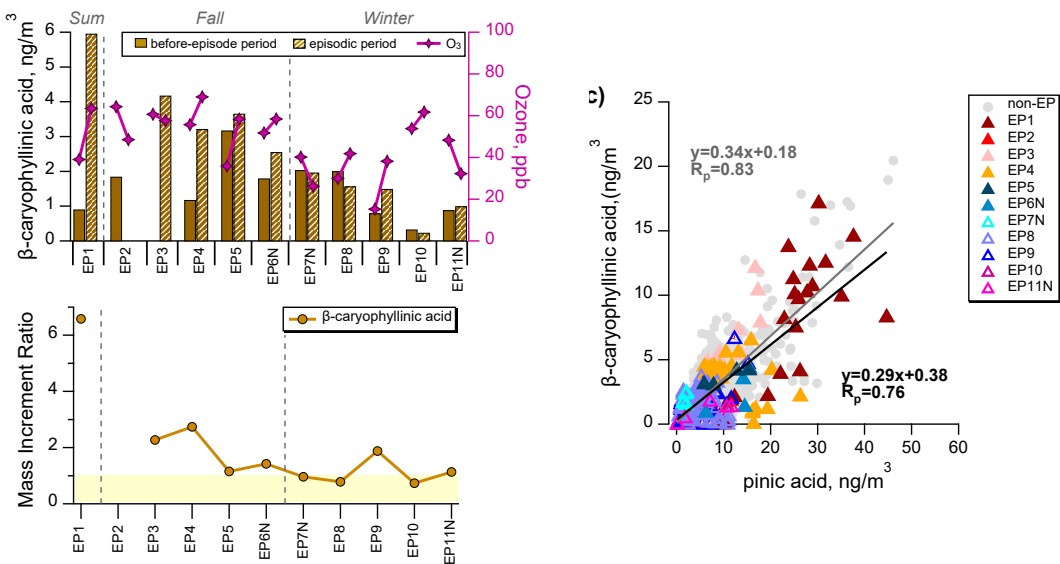


**Figure 7. (a) Comparison of average concentration of β-caryophyllinic acid in the before-episode periods (solid filling) and episodic periods (pattern filling). (b) Mass increment ratio of β-caryophyllinic acid for individual episodes, with the light-yellow area making the ratio values less than 1. (c) Correlation between β-caryophyllinic acid and pinic acid in the episodes and non-episodic hours.**

## 4 Conclusions

Detailed online PM$_{2.5}$ speciation measurements including major inorganic ions, OC, EC, elements, and organic molecular markers were conducted at a suburban site over a four-month campaign from 30 Aug. to 31 Dec. 2020, spanning over three seasons (summer, fall, and winter). Taking advantages of the hourly/bihourly chemical composition data, especially precursor-specific SOA tracers, we examined the evolution of SOA tracers at this site during eleven city-wide PM$_{2.5}$ episodes falling in our measurement period. The PM$_{2.5}$ episodes were identified based on PM$_{2.5}$ data from a network of

15 air quality monitoring stations across the whole city. They were distributed in three seasons, with one in summer, five





in fall, and five in winter. PM$_{2.5}$ in episodes in summer and early fall showed less spatial variation compared with the winter episodes. Among the SOA tracer groups, notably lower concentrations in winter were observed for two groups of biogenic SOA tracers (i.e., those derived from isoprene and monoterpene) and the monoaromatic SOA tracer. Biomass burning POA and SOA tracers (i.e., levoglucosan and 4-nitrocatechol, respectively) and the naphthalene/methylnaphthalene SOA tracer (i.e., phthalic acid) showed higher concentrations in winter and fall. Mass increment ratios, calculated as the ratio between before- and during-episode concentrations, mostly exceeded 1 for individual groups of SOA tracers, indicating enhanced SOA formation during episodes. The maximum MIR value encountered was 11 for the isoprene SOA tracers and similar for other groups of SOA tracers (4.2-7.0), demonstrating the significant potential for SOA to contribute to PM$_{2.5}$ episodic pollution. The MIR values of SOA tracers were generally higher during the summer/early fall episodes while lower during the winter episodes, implying SOA formation more sensitive to the oxidant level in summer and fall while more sensitive to the VOC precursors in winter.

Multiple SOA tracers are available for isoprene and monoterpene SOA, providing an opportunity to gain insights into their formation mechanism. Among the six isoprene SOA tracers, C$_5$-alkene triols and 2-methyl tetrols consistently dominated over 2-methylglyceric acid. This observation showed the importance of low-NO$_x$ formation pathways in isoprene SOA. Among the monoterpene SOA tracers, the relative abundance of pinic acid and 3-MBTCA (the P/M ratio) indicated the dominance of the early generation products in winter and SOA are generally less aged in winter. Future efforts are recommended to direct at laboratory identification of multiple products from a single SOA precursor, preferably representing products from different pathways or oxidation stages. Their joint field monitoring will greatly facilitate to develop quantitative understanding of SOA formation under real-world conditions.

The current study has shown the SOA chemical evolution characteristics during PM$_{2.5}$ episodes vary by precursors and by seasons. While we have demonstrated the value of online monitoring of specific molecular tracers in tracking episodic events and in examining episode-scale SOA formation characteristics, instrumentation at one site is insufficient to adequately capture the spatial heterogeneity of the haze pollution at the city scale. For formulating PM control measurements specific to a city or a region, multiple-site monitoring with advanced online instruments is highly recommended.

*Data availability.* Bihourly organic markers and other hourly chemical speciation data presented in this study can be requested by contacting the corresponding author (jian.yu@ust.hk).

*Author contribution.* QW and JZY formulated the overall design of the study. QW and SW carried out the measurement of organic markers and data validation. YYC, HC, ZZ, DG, ZW and JL carried out the measurement of other key major



components and data validation. QW analyzed the data with contributions from JZY. QW and JZY prepared the manuscript
with contributions from all co-authors.

*Competing interests.* The authors declare that they have no conflict of interest.

*Disclaimer.* The content of this paper does not necessarily reflect the views and policies of the HKSAR Government,
nor does mention of trade names or commercial products constitute an endorsement or recommendation of their use.

*Acknowledgements.*    We thank funding support from the Hong Kong Research Grants Council (R6011-18 and
16305418), and the Hong Kong University of Science & Technology (VPRDO19IP01).

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
