# Peer review of "Chemical evolution of secondary organic aerosol tracers during high $PM_{2.5}$ episodes at a suburban site in Hong Kong over 4 months of continuous measurement"

_Atmospheric Chemistry and Physics, 2022_

## Author Comment (AC1)

*Response to Review Comments by Anonymous Referee #1 on "Chemical evolution of secondary organic aerosol tracers during high PM2.5 episodes at a suburban site in Hong Kong over 4 months of continuous measurement" by Q. Wang et al.*

**General Comments by Anonymous Referee #1:**

This manuscript reported a four-month's continuous bihourly measurement of various types of secondary organic aerosol (SOA) tracers originated from common specific volatile organic compounds (VOCs). The authors focused on the concentration change of the SOA tracers during high $PM_{2.5}$ episodes and found obvious mass increment of the SOA tracers during the episodes especially in summer and fall, suggesting enhanced SOA formation during the episode. Moreover, the chemical formation mechanism and ageing of the biogenic SOA formation during the episodes were examined with the measurement of multiple molecular tracers from single VOC precursor, which provide valuable insight into the formation mechanism under ambient atmosphere similar to this study area. In general, this manuscript is well written, and the dataset is large. The manuscript can be accepted with the following revisions:

**Response to General Comments:** We thank the reviewer for the comments and acknowledgment of the importance of our work. Below is our point-by-point response to each comment, marked in blue. Changes to be made to the main text are also marked in blue in the revised manuscript file.

1.  As the focus of this work is to examine the SOA tracers, while most of which lack authentic standards for the quantification. The authors should elaborate more on how the identification and quantification was achieved (such as the retention time, quantification ions, etc.).

    **Response:** The measurement of individual SOA tracers via derivatization followed by GC/MS method have been well documented in previous studies. The mass spectra of the SOA tracers studied in this work have been reported in previous studies, for example, Claeys et al. (2004) for isoprene SOA tracers, Szmigielski et al. (2007) for α-pinene SOA tracers, Jaoui et al. (2007) for β-caryophyllinic acid, and Al-Naiema and Stone, (2017) for DHOPA. In this work, the SOA tracer species without authentic standards were identified by comparing their ambient mass spectra with the previously reported data and quantified by using surrogate compounds with similar structures and functional groups. We've added the retention time and quantification ions for all the measured organic markers in Figure S1.

    **Lines 106-111:** "Levoglucosan, 4-nitrocatechol, phthalic acid and pinonic acid, with available authentic standards, were identified and quantified by directly comparing to their standards. The remaining SOA tracers which do not have authentic standards were identified by comparing their ambient mass spectra with the previously reported data (Al-Naiema and Stone, 2017; Claeys et al., 2004; Jaoui et al., 2007; Szmigielski et al., 2007) and quantified using the surrogate compounds with similar structures and functional groups. The retention time and quantification ions of each species are shown in Figure S1."

[Figure]

**Figure S1.** Example of the extracted ion chromatograms from select ambient sample (4 Sep. 2020 4:00 am) for the target organic markers studied in this work.

2. The SOA tracers are widely used to estimate the SOC contribution either by the SOA tracer method or by receptor models such as Positive Matrix Factorization (PMF). Since the comprehensive measurement data are available, do the authors plan to conduct such analysis to get a more quantitative estimation of the SOA contribution?

   **Response:** Yes, we have a plan to conduct a more quantitative source analysis using PMF of a longer dataset and this will be presented in a future publication.

3. Lines 190-194: Why the authors only selected the "before-episode period" to evaluate the evolution of SOA tracers, instead of using all measured data?

   **Response:** The campaign-wide average is not representative to the normal condition specific to individual seasons, as the emission sources and meteorological conditions varied among seasons. The selection of the "before-episode period" can minimize the interference from the different meteorological conditions such as temperature and boundary layer height among seasons. This comparison (i.e., before-episode period vs. episodic period) can better examine the rapid formation of the high PM episodes.

4. The date format in the main text and Tables is not consistent. For example, "10 Jul. -7 Oct" in Line 135 vs. "Sep-01 12:00 PM" in Table 1. Please check and unify the format.

   **Response:** the time format in Table 1 and Table S1 and elsewhere have been revised accordingly. Please see the updated Table 1.

**Table 1.** Statistical summary of PM$_{2.5}$ at 15 HKEPD air quality monitoring stations and the HKUST supersite during the 11 episodes and the remaining non-episodic hours during the period of 10 Jul.-31 Dec. 2020. Meteorological parameters and gas pollutant data are from HKUST supersite.

| Episodes | Season | Time period | Duration (h) | Wind speed (m s$^{-1}$) | T (oC) | RH (%) | O$_3$ (ppb) | NO$_x$ (ppb) | Avg. PM$_{2.5}$, µg m$^{-3}$ | | | |
|---|---|---|---|---|---|---|---|---|---|---|---|---|
| | | | | | | | | | City-wide avg. | Min. | Max. | Max-to-Min ratio |
| EP1 | Summer | 1 Sep 12:00 PM – 4 Sep 3:00 PM | 76 | 1.33 | 29.7 | 78.0 | 63.4 | / | 37.6 | 32.1 | 44.2 | 1.4 |
| EP2 | Fall | 30 Oct 7:00 AM - 7:00 PM | 13 | 2.72 | 23.2 | 79.0 | 48.5 | 12.6 | 32.6 | 26.1 | 39.9 | 1.5 |
| EP3 | Fall | 2 Nov 7:00 AM – 4 Nov 9:00 PM | 63 | 3.93 | 23.2 | 65.5 | 57.6 | 12.0 | 30.7 | 26.6 | 37.8 | 1.4 |
| EP4 | Fall | 6 Nov 11:00 AM – 10 Nov 10:00 PM | 108 | 3.36 | 24.2 | 56.9 | 69.0 | 13.1 | 33.0 | 28.4 | 41.0 | 1.4 |
| EP5 | Fall | 24 Nov 12:00 PM - 7:00 PM | 8 | 3.44 | 22.4 | 79.1 | 58.4 | 8.22 | 29.7 | 22.1 | 43.0 | 1.9 |
| EP6N | Fall | 26 Nov 4:00 PM – 27 Nov 1:00 AM | 10 | 1.36 | 20.8 | 89.6 | 58.5 | 7.20 | 29.9 | 17.2 | 44.9 | 2.6 |
| EP7N | Winter | 3 Dec 1:00 AM - 10:00 AM | 10 | 6.13 | 15.6 | 67.7 | 26.2 | 11.2 | 31.0 | 24.3 | 38.9 | 1.6 |
| EP8 | Winter | 5 Dec 2:00 AM – 13 Dec 0:00 AM | 191 | 2.59 | 18.7 | 71.9 | 41.8 | 13.9 | 33.2 | 25.6 | 47.9 | 1.9 |
| EP9 | Winter | 19 Dec 1:00 PM – 25 Dec 10:00 PM | 154 | 3.62 | 16.3 | 68.4 | 38.2 | 11.3 | 28.4 | 20.6 | 41.9 | 2.0 |
| EP10 | Winter | 27 Dec 11:00 AM – 28 Dec 11:00 AM | 25 | 1.89 | 20.5 | 58.0 | 61.7 | 6.69 | 28.9 | 15.7 | 66.7 | 4.3 |
| EP11N | Winter | 29 Dec 8:00 PM – 30 Dec 4:00 AM | 9 | 3.70 | 18.5 | 72.1 | 32.2 | 21.3 | 33.8 | 22.8 | 43.4 | 1.9 |
| non-EP | / | 10 Jul 0:00 AM – 31 Dec 11:00 PM | 3533 | 2.88 | 25.0 | 78.9 | 44.8 | 9.10 | 12.5 | 10.4 | 16.9 | 1.6 |

5. Figure 1: there are only two sites showing numbers near the column plot in the figure, are they the concentration values stated in the figure caption? How about the other sites?

**Response:** Yes, the numbers indicate the average PM$_{2.5}$ concentrations during the non-episodic period for each site (Figure 1). Previously, when converting the word file to pdf files, the numbers somehow were erased. We will fix the problem in the revised manuscript.

[Figure]

**Figure 1.** Geographical location of the 15 HKEPD general air quality monitoring stations (black dot) and the HKUST supersite (red star) in HK. Among the 15 HKEPD stations, MB is the rural site while others are general urban stations with different microenvironments. Column plot shows the average PM$_{2.5}$ concentrations (µg m$^{-3}$) during non-episodic hours (in black, with numbers indicating the concentration values) and the eleven episodes (in various colors).

6.   Figures 5-6: the figure numbers "(a)" and "(b)" in the figure are missing.

**Response:** The figure numbers will be added.

7.   Figure S8: The axis of Figure S8 is not readable.

**Response:** The revised Figure S8 (now become Figure S9) is copied below.

[Figure]

**Figure S9:** Diurnal variations of (a) levoglucosan, 4-nitrocatechol, DHOPA, phthalic acid, and β-caryophyllinic acid; (b) individual α-pinene SOA tracers; (c) individual isoprene SOA tracers, and (d) $O_3$, $NO_x$ and $PM_{2.5}$ under the non-episodic and episodic period from 30/8-31/12, 2020 at HKUST supersite.

8.   Line 193: "were" should be "are".

**Response:** Suggestion taken.

9.   Line 230: "in Chow et al. 2016)." should be "in Chow et al. (2016)).".

**Response:** Suggestion taken.

10.  Lines 297-298: The description is not consistent with the content of Figure 5. At EP6N, isoprene SOA tracer concentrations were higher in the time period before the episode.

**Response:** we have revised the statement:

**Lines 306-307:** "The isoprene SOA tracers were consistently higher during episodes than the before-episode periods except for one nighttime episode-EP6N (Figure 5)."

11. Lines 374-379: The description is not consistent with Figure 7.

    **Response:** The reviewer may have misunderstood the statement. Here we want to discuss the less decrease in concentration in winter compared with summer and fall for β-caryophyllinic acid (i.e., less seasonal contrast). We've re-phrased the statement as following:

    **Lines 383-385: "**It is plausible that the less decrease in concentration of β-caryophyllinic acid in both non-episodic and episodic periods in winter compared with those in summer and fall could be a result of the lower atmospheric oxidative capacity in winter leading to less degradation of β-caryophyllinic acid."

12. Line 403: "2-methyl tetrols" should be "2-methyltetrols".

    **Response:** revised as suggested.

---

## Author Comment (AC2)

*Response to Community Comments by Referee #2 on "Chemical evolution of secondary organic aerosol tracers during high PM2.5 episodes at a suburban site in Hong Kong over 4 months of continuous measurement" by Q. Wang et al.*

**General Community Comments** *by Referee #2*:

The current study has been designed to monitor SOA tracers at a suburban site in Hong Kong for four months. Results showed regional characteristics for anthropogenic and biogenic SOA including for biomass burning SOA. This study also highlights the need of high time resolution organic marker measurement at multiple sites to fully capture the spatial variability and implement control measures. I think such kind of field study should be promoted in future to completely understand the role of SOA formation during haze events.

**Response to General Comments:** We thank the reviewer for the comments and affirming the importance of our work, which provided the molecular-level evidence of the enhanced SOA formation during the episodic hours in different seasons.

Below is our point-by-point response to each comment, marked in blue. Changes to be made to the main text are also marked in blue in the revised manuscript file.

My suggestions are given below:

1.  Please add reference "PM pollution was observed to have a clear seasonal pattern, with lower concentrations in summer and higher in fall and winter."

    **Response:** Suggestion taken, the following reference was added:

    **Lines 41-42:** "PM pollution was observed to have a clear seasonal pattern, with lower concentrations in summer and higher in fall and winter (Huang et al., 2014)."

    Huang, X. H. H., Bian, Q., Ng, W. M., Louie, P. K. K. and Yu, J. Z.: Characterization of PM2.5 major components and source investigation in suburban Hong Kong: A one year monitoring study, Aerosol Air Qual. Res., 14(1), 237–250, doi:10.4209/aaqr.2013.01.0020, 2014.

Please add reference" In this work, we define that a PM2.5 pollution episode occurred when the PM2.5 concentration was higher than 35 μg/m$^3$ (24-hour standard)"

    **Response:** Suggestion taken. The following reference was added:

    **Lines 143-145:** "In this work, we define that a $PM_{2.5}$ pollution episode occurred when the $PM_{2.5}$ concentration was higher than 35 μg m$^{-3}$ for at least consecutive six hours at three or more stations. This value (35 μg m$^{-3}$) is the current annual $PM_{2.5}$ air quality objective by the Hong Kong government, which aligns with the WHO's interim target-1 value for annual $PM_{2.5}$ (WHO, 2005)."

    WHO: World Health Organization. Regional Office for Europe. (2006). Air quality guidelines: global update 2005: particulate matter, ozone, nitrogen dioxide and sulfur dioxide. World Health Organization, Regional Office for Europe, 2005

2.  Line 220: What about levoglucosan and nitrocatechol correlation in different periods? Have authors also checked other meteorological parameters? Sometimes meteorology could affect the existing correlation.

    **Response:** Yes, we've examined the correlation between 4-nitrocatechol and levoglucosan, and the meteorological factors including temperature, relative humidity and wind speed. They did not affect the moderate correlation feature between the two species (Figure R1).

[Figure]

**Figure R1.** Scatter plots between 4-nitrocatechol and levoglucosan, color coded by temperature, relative humidity (RH) and wind speed.

3. Line 230: "This likely reflects that 4-nitrocatechol has precursor sources other than BB and joint measurements of potential precursors (e.g., catechol, phenol, benzene) in the future would help to discern the relative importance of precursors from BB versus anthropogenic sources." Yes, nitrocatechol has other precursors and author should cite those references here.

**Response:** Suggestion taken, the following reference was added:

**Lines 236-238:** "This likely reflects that 4-nitrocatechol has precursor sources other than BB (Lu et al., 2019; Yuan et al., 2021) and joint measurements of potential precursors (e.g., catechol, phenol, benzene) in the future would help to discern the relative importance of precursors from BB versus anthropogenic sources."

Lu, C., Wang, X., Li, R., Gu, R., Zhang, Y., Li, W., Gao, R., Chen, B., Xue, L. and Wang, W.: Emissions of fine particulate nitrated phenols from residential coal combustion in China, Atmos. Environ., 203, 10–17, doi:10.1016/j.atmosenv.2019.01.047, 2019.
Yuan, W., Huang, R. J., Yang, L., Wang, T., Duan, J., Guo, J., Ni, H., Chen, Y., Chen, Q., Li, Y., Dusek, U., O'Dowd, C. and Hoffmann, T.: Measurement report: PM2.5-bound nitrated aromatic compounds in Xi'an, Northwest China - Seasonal variations and contributions to optical properties of brown carbon, Atmos. Chem. Phys., 21(5), 3685–3697, doi:10.5194/acp-21-3685-2021, 2021.

Have author checked nitrocatechol to levoglucosan ratio for different season?

**Response:** The 4-nitrocatechol to levoglucosan ratio is shown in Figure R2. Generally, higher ratios were observed in fall and winter period. The 4-nitrocatechol/levoglucosan ratio from different sources were not available from the literature, thus it is difficult to compare the observational data to the source profiles.

[Figure]

**Figure R2.** Seasonal variation of the 4-nitrocatechol/levoglucosan ratio in this study.

4. Line 260: Did author suggest the role of ozone in the formation of phthalic acid and DHOPA? I will appreciate if more information can be provided on their formation pathways.

   **Response:** The following will be added to the revised manuscript for clarification and elaboration on the point raised by the reviewer.

   **Lines 270-273:** "Note that we used ozone as an indicator for the oxidant level in the ambient atmosphere, as no measurements of OH radical were available. The formation pathways for phthalic acid and DHOPA are mostly via OH radical oxidation, as reported in previous studies (He et al., 2018; Wang et al., 2007; Zhang et al., 2021)."

He, X., Huang, X. H., Chow, K. S., Wang, Q., Zhang, T., Wu, D., & Yu, J. Z. (2018). Abundance and sources of phthalic acids, benzene-tricarboxylic acids, and phenolic acids in PM2. 5 at urban and suburban sites in Southern China. *ACS Earth and Space Chemistry*, 2(2), 147-158.

Wang, L., Atkinson, R., & Arey, J. (2007). Dicarbonyl products of the OH radical-initiated reactions of naphthalene and the C1-and C2-alkylnaphthalenes. *Environmental Science and Technology*, 41(8), 2803-2810.

Zhang, J., He, X., Gao, Y., Zhu, S., Jing, S., Wang, H., ... & Ying, Q. (2021). Estimation of aromatic secondary organic aerosol using a molecular tracer—a chemical transport model assessment. Environmental Science & Technology, 55(19), 12882-12892.

---

## Author Response (AR2)

*Response to Review Comments by Editor on "Chemical evolution of secondary organic aerosol tracers during high PM2.5 episodes at a suburban site in Hong Kong over 4 months of continuous measurement" by Q. Wang et al.*

**Comments to the author:**

The following two responses to the referee's comments should be included in the revised manuscript, because they comprise relevant information for readers. Figure R1 could be included in the supplement if authors consider so.

**Response to General Comments:** We thank the editor for the comments. We've included the two responses in the revised manuscript. Below is our point-by-point response to each comment, marked in blue. Changes made to the main text are also marked in blue in the revised manuscript file.

REFEREE # 1

3. Lines 190-194: Why the authors only selected the "before-episode period" to evaluate the evolution of SOA tracers, instead of using all measured data?

Response: The campaign-wide average is not representative to the normal condition specific to individual seasons, as the emission sources and meteorological conditions varied among seasons. The selection of the "before-episode period" can minimize the interference from the different meteorological conditions such as temperature and boundary layer height among seasons. This comparison (i.e., before-episode period vs. episodic period) can better examine the rapid formation of the high PM episodes.

**Response:** suggestion taken; the following statement has been added into the main context in **Lines 194-198**:

"To characterize chemical features in the formation of PM episodes, we examined the PM composition before and during each episode. The campaign-wide average is not representative to the normal condition specific to individual seasons, as the emission sources and meteorological conditions varied among seasons. The selection of the "before-episode period" can minimize the interference from the different meteorological conditions such as temperature and boundary layer height among seasons. This comparison (i.e., before-episode period vs. episodic period) can better examine the rapid formation of the high PM episodes."

REFEREE # 2

2. Line 220: What about levoglucosan and nitrocatechol correlation in different periods? Have authors also checked other meteorological parameters? Sometimes meteorology could affect the existing correlation.

Response: Yes, we've examined the correlation between 4-nitrocatechol and levoglucosan, and the meteorological factors including temperature, relative humidity and wind speed. They did not affect the moderate correlation feature between the two species (Figure R1).

**Response:** suggestion taken; the following statement has been added into the main context in **Lines 229-231**:

"In our dataset, a moderate correlation of 4-nitrocatechol with levoglucosan was observed ($R_p$: 0.43), in line with their common material origin from BB. The moderate correlation feature between the two species was not affected by the meteorological factors (Figure S14)."

[Figure]

**Figure S14.** Scatter plots between 4-nitrocatechol and levoglucosan, color coded by temperature, relative humidity (RH) and wind speed.